# Two-Step Electrochemical Au Nanoparticle Formation in Polyaniline

**DOI:** 10.3390/nano13142089

**Published:** 2023-07-17

**Authors:** Bin Zhao, Hans-Werner Becker, Sebastian Gutsch

**Affiliations:** 1IMTEK, Faculty of Engineering, Albert-Ludwigs University Freiburg, Georges-Köhler-Allee 103, 79110 Freiburg, Germany; bin.zhao@imtek.uni-freiburg.de; 2RUBION, Zentrale Einrichtung für Ionenstrahlen und Radionuklide, Ruhr-Universität Bochum Universitätsstr, 150, 44780 Bochum, Germany; hans-werner.becker@rub.de

**Keywords:** polyaniline, gold nanoparticles, electrochemistry, TEM, RBS

## Abstract

In this work, we use a two-step cyclic electrochemical process to insert Au into polyaniline (PANI). It was suggested previously that this method would lead to the formation of atomic Au clusters with controlleds number of Au atoms without providing morphological proof. In each cycle, tetrachloroaurate anions (AuCl_4_^−^) are attached on the protonated imine sites of PANI, followed by a controlled reduction using cyclic voltammetry (CV). In contrast to previous work, we demonstrate that the reduction leads to the nucleation and growth of an Au nanoparticle (NP) whose density and size dispersion depend on the Au loading in PANI. Adding more deposition cycles increases the Au NP density and size. Transmission electron microscopy (TEM) and corresponding energy dispersive X-ray spectroscopy (EDS) indicate a homogeneous distribution of Au elements in the PANI matrix before CV reduction, while Au elements are aggregated and clearly localized in the NPs positions after CV reduction. We further use Rutherford backscattering spectrometry (RBS) to quantify the Au uptake in PANI. The Au distribution is verified to be initially homogeneous across the PANI layer whereas the increasing number of deposition cycles leads to a surface segregation of Au. We propose a two-step growth model based on our experimental results. Finally, we discuss the results with respect to the formation of atomic Au clusters reported previously using the same deposition method.

## 1. Introduction

Among the conducting polymers, PANI has recently gained interest due to its unique electrochemical properties [1]. PANI is one of the most widely studied polymers as it possesses environmental stability, controllable high conductivity and redox properties [2]. In particular, applications of PANI/Au composites have been widely explored in electrochemical catalysis [3], chemical gas sensing [4], fuel cells [5] and biosensors [6]. Electrochemical introduction of noble metals such as Au, Pt and Pd into PANI has been previously studied [7,8,9,10]. In these works, noble metal halide anions are reduced by PANI during an electrochemical potential sweep leading to the formation of a variety of PANI/noble NP heterostructures. Even though many experimental parameters can be tuned to achieve variations with respect to NP size and metal loading, the noble metal deposition still follows the laws of thermodynamics leading to a significant NP size dispersion.

A substantial advance of the electrochemical introduction of Au in PANI was achieved by Janata and co-workers, who used a two-step cyclic electrochemical process to insert one atom per site and cycle [11,12]. The process takes advantage of the high affinity of AuCl_4_^−^ with the protonated imine sites of PANI. The insertion is carried out in an electrochemical flow cell. First, the PANI electrode is held at a voltage of +0.8 V vs. Ag/AgCl and then a solution of dilute AuCl_4_^−^ precursor is introduced into the flow cell and presumably attaches to the said imine sites. This step is followed by excessive rinsing while still holding the electrochemical potential in order to remove any unbound AuCl_4_^−^. Finally, the remaining bound AuCl_4_^−^ is reduced by sweeping the electrode potential to −0.2 V vs. Ag/AgCl which is believed to lead to a controlled on-site reduction. This process can be repeated *n* number of times, effectively adding a single Au atom per site. While the exact details of the AuCl_4_^−^ binding and reduction process are not known, it was demonstrated that electrodes prepared in this way have catalytic properties that follow the odd-even pattern of atomic Au clusters [11]. This idea is substantiated by a concomitant odd-even energy oscillation of the NH-related infrared absorption band. The work was later expanded to the formation of atomic Pd clusters and mixed Au/Pd clusters [13]. The latter even showed electrochemical activity towards the oxidation of alcohols with respect to their order of deposition [14]. Recently, these results were backed up by density functional theory calculations [15]. It was shown that such triatomic clusters of Au and Pd are indeed stabilized on the quinone ring by a large interaction energy. This result is significant as it also gives a hint on the working principle of sequential atomic metal build-up in PANI. After metal reduction, the metal localizes on the quinone ring, effectively freeing up the imine site for subsequent insertion cycles.

Many open questions still remain regarding the details of the atomic metal insertion process. For instance, nothing is known to date on the influence of cyclic AuCl_4_^−^ precursor dose on the creation, density, distribution and stability of the atomic metal species. It was shown that in case of electroless exposure of AuCl_4_^−^ to PANI, more than five times Au can be absorbed by PANI with respect to its weight [16]. However, given a limited number of active nucleation sites in the atomic metal insertion process, it is reasonable to expect saturation at lower ratios. 

In this article, we investigate the introduction of Au species into PANI via precise electrochemical control in a flow cell system. PANI/Au composites produced in this way are characterized using scanning electron microscopy (SEM), TEM and RBS to quantify the amount of introduced Au for the first time. It will be demonstrated that the PANI/AuCl_4_^−^ complex is stable prior to the reduction process. However, in contrast to the work of Janata et al. [11,12,15], we do not see any signs of atomic Au species after the controlled reduction of these PANI/AuCl_4_^−^ complexes. Instead, we do clearly observe the formation of Au NPs. 

## 2. Materials and Methods

### 2.1. Synthesis of Polyaniline Films

Glassy carbon (GC) disks (13 mm diameter, 1 mm thick, HTW, Thierhaupten, Germany) with resistivity of 30.2 µOhm⋅cm and bulk density of 1.50 g/cm^3^ were used as substrates. PANI was deposited using a Gamry Interface 1010 potentiostat in a three-electrode configuration. This electrochemical system contains the GC substrate functioning as a working electrode (WE), one Ag/AgCl in 3 M NaCl/3 M NaNO_3_ self-made reference electrode (RE) and a platinum foil as counter electrode (CE). PANI was electrochemically polymerized in a 0.1 M aniline/2 M HBF_4_ aqueous solution in a beaker. First, a linear sweep voltammetry was performed from open circuit potential (OCP) to 1 V with a scan rate of 100 mV/s. Afterwards, electrodeposition was carried out by chronoamperometry at a constant potential of 0.90 V until a total charge of 400 mC was reached. Ion exchange of the grown PANI films was finally completed in a 0.1 M HClO_4_ solution within the potential ranging from −0.2 V to +0.85 V with a scan rate of 50 mV/s for 3 cycles followed by drying in a vacuum.

### 2.2. Atomic Gold Deposition in Polyaniline

The whole experimental setup is shown in Figure 1a. In contrast to previous work, we used a platinum tube as CE at the flow cell outlet [17,18,19]. The Au insertion process requires a continuous control of the applied potential on PANI during the AuCl_4_^−^ exposure. To this end, a cross-flow cell (Figure 1b) was designed, where the RE (screw type RE-3VT, ALS) is placed directly above the WE. The fluidic setup further consisted of two peristaltic micropumps (model 400FD, Watson Marlow), a chemical inert 4-port, 2-position valve (Vici Valco) and two solution reservoirs as well as a waste reservoir. All fluidic connections were made using teflon tubes. A compact potentiostat was used for the electrochemical control (Emstat3+, PalmSens, Houten, Netherlands). All hardware was controlled by a C# program which also allows to run fully automated deposition sequences. 

After mounting the PANI/GC samples in the flow cell, the whole setup was primed with 0.1 M HClO_4_ solution. The method of Au deposition adopted from Jonke et al. [12] is shown in Figure 1c. First, the applied potential is increased from OCP to +0.8 V (Stage I to Stage II). At +0.8 V, PANI is oxidized from PANI(ES) to PANI(PNS), i.e., the amine sites in PANI(ES) convert to imine in PANI(PNS). The sample is then exposed to 0.1 mM KAuCl_4_/0.1 M HClO_4_ solution while holding the potential at +0.8 V (Stage III). The high affinity of AuCl_4_^−^ with the imine sites is believed to form bound complexes. Afterwards, the excess AuCl_4_^−^ is rinsed away by 0.1 M HClO_4_ solution while still holding the potential at +0.8 V. This is followed by a CV sweeping back to −0.2 V to reduce PANI(PNS)*AuCl_4_^−^ complex into Au(0) and PANI(ES) (Stage IV). The reduction process is supposed to free up the amine sites in PANI(ES), rendering them available again for the following cycle of PANI(PNS)*AuCl_4_^−^ complexation (Stage V). In contrast to previously described synthesis processes of PANI/Au composite structures, this method of Au deposition allows to separately address the exposure of AuCl_4_^−^ and the reduction process individually by preventing the reduction during AuCl_4_^−^ exposure by means of electrochemical potential control while also getting rid of excess AuCl_4_^−^ by rinsing excessively.

The detailed timing diagram of the stage transitions from I to V including the experimental parameters is shown in Figure 2 and Table 1 for the Au deposition process in PANI. Four different sample groups erre prepared. In group A, the exposure time (t_E_) of AuCl_4_^−^ was varied from 1 min to 16 min without any intentional CV reduction step. In contrast, group B represents samples of a single complete deposition cycle with different exposure times ranging from 1 min to 16 min. Furthermore, a total sequence ranging from 1 to 6 cycles was achieved with a constant exposure time of 1 min for each cycle (group C). An additional single cycle deposition with 1 mM KAuCl_4_ solution was performed to evaluate the role of the precursor concentration (sample D).

### 2.3. Characterization

The Au deposited PANI samples were imaged by a Nova NanoSEM 430 (Thermo Fisher Scientific, Waltham, MA, USA) scanning electron microscope at a voltage of 10 kV to investigate the sample morphology. Firstly, 10 images were taken for each sample from different positions and henceforth evaluated in order to analyze the size dispersion of Au NPs by ImageJ [20]. Before TEM observation, the sample surface was scratched and carefully peeled off with a stainless steel blade. The peeled-off powders were then transferred onto a holey carbon TEM grid. The holder was put into a Talos 120C (Thermo Fisher Scientific, Waltham, MA, USA) transmission electron microscopy for sample morphology observation and structure measurement at an acceleration voltage of 200 kV. HAADF-STEM in conjunction with EDS was used for the elemental mapping. The elemental concentration of Au deposited PANI was measured using RBS (RUBION, Bochum, Germany). RBS allows elemental quantification without the use of reference or calibration standards, while most other techniques use a correction factor or need references of known composition to determine the concentrations. The RBS measurement was performed using 2 MeV He(4) projectiles at a dose of 10 µC. The solid angle and detection angle in the Cornell geometry were 1.91 msrad and 160 degrees, respectively. The established software package RBX5.40 was used to directly obtain the Au concentrations from the integrated Au peak counts [21]. Peak fits were performed using Origin to fit overlapping O and N peaks. 

The reduction CVs for multiple deposition cycles are plotted in Figure 3, including the original PANI (short dot line), the first reduction CV sweep after holding the potential at +0.8 V (solid line) and the subsequent stable CV (dash line). The first reduction CV sweep corresponds to the reduction of the AuCl_4_^−^ anions of the PANI(PNS)*AuCl_4_^−^ complex [12]. It is also associated with the release of ions from PANI after holding at a high potential. The stable CV is characteristic for PANI/Au in acid solution. A negative shift in contrast to the original PANI is observed due to PANI degradation, resulting in a flat and featureless CV for Au [6c, 1 min, CV] (sample C6). 

## 3. Results and Discussion

### 3.1. Morphology of PANI Films with Inserted Au

The corresponding representative SEM images of the samples are shown in Figure 4. The samples exhibit the typical nanofibrillar PANI morphology and show the typical electrochemical behavior as observed in literature [22,23,24]. For a single cycle of Au deposition without any intentional reduction by CV (group A), no NPs are observed with the increase of exposure time from 1 min to 16 min, as shown in Figure 4a,b. A similar result is observed in Au [1c, 1 min, CV] (sample B1), where exposure time is also 1 min albeit it is followed by an additional CV for AuCl_4_^−^ reduction. For Au [1c, 2 min, CV] (sample B2, Figure 4d), a small amount of isolated small NPs in the range of about 5 nm can be found. As the exposure time increases, the density of NPs increases and larger aggregates of up to several hundred nanometers can be observed, as indicated in Figure 4e–h. Clearly, compared with group A, the reduction using the CV results in the nucleation and aggregation of Au NPs in group B. A similar phenomenon is observed for group C when the number of Au deposition cycles increases (Figure 4i–n). As can be seen in Au [2c, 1 min, CV] (sample C2, Figure 4j), NPs already start to appear clearly when the exposure time is only 1 min for each cycle. For a total net exposure time of 2 min, only rare occurrences of NPs are found in both Au [1c, 2 min, CV] and Au [2c, 1 min, CV] respectively. With the stepwise increase of Au deposition cycles, the density and size of NPs grow (Figure 4k–n). It appears that in group C (cycle number variation), the NPs are more finely dispersed and have a higher density than in group B (single reduction step). This indicates that each insertion cycle creates new NPs. Compared with Au [1c, 1 min, CV], a tenfold increase of the AuCl_4_^−^ concentration already leads to the appearance of NPs in sample D after a single full deposition cycle (Figure 4o). It is interesting to note that after the He ion bombardment in the RBS measurement, extremely tiny NPs (2–4 nm) are found in the measurement spot of Au [1c, 16 min, no CV] (Figure 4p). It indicates that Au was initially well-dispersed in PANI, but the high He ion energy resulted in the formation of Au NPs significantly smaller than the CV reduction in solution.

### 3.2. Size Distribution of Au NPs in PANI Films

As can be seen from SEM pictures, we could not detect NPs in group A within the resolution of the SEM. Further supportive TEM analysis is presented in the following section. In group B, the extremely low density and irregular shape of the NPs render any analysis on size distribution challenging. Therefore, we only took a closer look at group C for samples that have been prepared with 3 or more cycles. The results are shown in Figure 5. It can be seen that the NP dispersion does not follow a continuous dispersion law. Interestingly, there seems to be a common region ranging from around 5 to 20 nm. The density of this portion of NPs significantly increases with the number of cycles. Compared with Au [3c, 1 min, CV] in Figure 5a, the already existing 5–20 nm, 25–30 nm and 40–50 nm regions still remain for Au [4c, 1 min, CV] in Figure 5b, and additional peak regions located at 60–70 nm and 75–80 nm are found. Furthermore, the different size peak regions continue to exist also in Au [5c, 1 min, CV] and Au [6c, 1 min, CV] in Figure 5c,d. The total density of NPs quantitatively increases from about 12 NPs/µm^2^ in Au [3c, 1 min, CV] to 34 NPs/µm^2^ in Au [6c, 1 min, CV]. 

### 3.3. TEM Analysis of PANI Films with Inserted Au 

TEM images of selected samples are shown in Figure 6. For Au [1c, 1 min, no CV] (Figure 6a), no NPs are found. When the Au exposure time reaches 16 min (Figure 6b), there are still no NPs, in agreement with the SEM observation. Compared with Au [1c, 1 min, no CV], the Au [1c, 1 min, CV] (Figure 6c) clearly shows the existence of NPs. This is not observed in the corresponding SEM observation (Figure 6c), which can be attributed to both the SEM resolution and contrast limit. Importantly, it implies that the CV reduction will lead to the formation of NPs. The presence of Au NPs in Au [1c, 1 min, CV] renders the existence of atomic gold in the chosen area unlikely. When the exposure time increases to 16 min, the CV reduction leads to the formation of aggregated NPs with dense density (Figure 6d). Huge NPs in the range of several hundred nanometers can be observed, in agreement with the SEM observation. From the HRTEM observation of Au [1c, 16 min, CV] (Figure 6e), the (poly)crystallinity of the Au nanoparticles with diameters of around 10 nm is found. The corresponding fast Fourier transform (FFT) indicates the (200) and (111) planes of Au.

To further investigate the Au distribution in the PANI/Au composites before and after CV reduction, the relevant EDS mappings of Au [1c, 16 min, no CV] and Au [1c, 16 min, CV] are shown in Figure 7 and Figure 8, respectively. It can be seen again in the HAADF-STEM image in Au [1c, 16 min, no CV] (Figure 7) that no NPs are found in the PANI matrix. Evidently, homogeneous distributions of C, N, O and Cl elements are found all across the composite. These elements originate from PANI and the HClO_4_ acid doping, respectively. In addition, Au shows a similar distribution compared with other elements, implying that Au is homogeneously distributed in the PANI matrix. For the reduced Au [1c, 16 min, CV] (Figure 8), many NPs are observed. The C, N and Cl elements are still homogeneously distributed in the PANI matrix while the Au elements are aggregated and clearly localized in the NP positions. 

### 3.4. Quantification of the Introduced Au 

RBS was measured for all samples as it allows to quantitatively determine the amount of Au deposited on or in the PANI film. The resulting spectra are very similar for all samples except for the Au-related peak. As an example, we plot in Figure 9a the spectra of two samples with the same net total exposure time of 4 min. Before we discuss the details, we first describe the general observations of all measured spectra. Two isolated peaks can be observed around channel 275 and channel 400, which can be assigned to Cl and Au, respectively. The peak area is directly proportional to a given elemental concentration, whereas the peak width is indicative of the elemental depth distribution. The low energy tailing of the peaks can be explained by the corrugated PANI surface as a result of the nanofibrillar morphology. As such, the apparent film thickness “seen” by the detector is not constant. Additionally, we can observe peaks for F, O, N and C at channels 180, 160, 140 and 110, respectively. However, there is a significant overlap of the peaks due to the film thickness and corrugation as well as the low difference in atomic number between these elements. Therefore, the elemental concentrations of N and O were determined in the following way. First, the Cl-peak was fitted with an asymmetric function. As the Cl is stemming mostly from the perchlorate anion which is needed for charge compensation in the PANI film, it is reasonable to assume that the Cl profile matches the PANI profile. Second, the same fitting parameters except for the scaling factor and translational variable were used for O and N. In this way, the true area of the latter elements could be determined and used for the elemental quantification, as shown in Figure 9b. The same procedure was also applied to fit the Au peak. From Figure 9a, it is apparent that the Au peak shapes for Au [1c, 4 min, CV] and Au [4c, 1 min, CV] are different. For Au [4c, 1 min, CV], a single steep peak of Au without a plateau region is observed. It indicates that the majority of deposited Au is located on the top surface of PANI. Instead, for Au [1c, 4 min, CV], the Au peak forms a plateau, indicating a more homogeneous depth distribution of Au. Please note that the plateau region is observed for all samples with just one deposition cycle. Interestingly, the Au peak can be fitted with the same fitting parameters as Cl (Figure 9b) for all one cycle samples, indicating a quite homogenous depth distribution of Au in the PANI film. However, as more deposition cycles are added, the Au peak gradually changes into a sharp peak which points towards surface aggregation of Au. 

Apart from Au, Cl, F, O and N have also been detected with the following average quantities across all measured samples: Cl = (1.12 ± 0.49) × 10^17^ atoms/cm^2^, F = (4.22 ± 3.76) × 10^15^ atoms/cm^2^, O = (3.06 ± 0.53) × 10^17^ atoms/cm^2^, and N = (4.29 ± 0.37) × 10^17^ atoms/cm^2^. The residual amounts of F must stem from the PANI deposition process which is carried out in HBF_4_. This detection indicates a somewhat incomplete ion exchange process. The Cl concentration seems to be very constant across all measured samples independent of the amount of AuCl_4_^−^ that was introduced. Next, the O/Cl ratio is ~2.7 which is lower than the expected ratio of ~4 for the perchlorate anion. This may indicate partial degradation of the perchlorate anion or incorporation of additional Cl ions possibly stemming from the reference electrode and/or the AuCl_4_^−^ reduction process. From the N concentration, we can estimate a “compacted” PANI layer thickness of ~540 nm, assuming a literature value of the volume density of PANI of 1.329 g/cm^3^ [25]. Please note that it is not possible to quantify the amount of C as the substrate itself consists of carbon. 

The measured Au elemental concentrations along with the determined Au/N ratio are shown in Figure 10. The latter is an interesting figure of merit to quantify the AuCl_4_^−^ sorption capability of PANI. The Au concentration versus exposure time indicates a linear Au uptake. For group B, it can be seen that Au concentration gradually increases from 0.6 × 10^15^ atoms/cm^2^ to 4.24 × 10^16^ atoms/cm^2^. A similar result is observed for group A where no CV reduction is applied. As no NPs are found in group A, the Au/N ratio can be used to quantitatively determine the percentage of imine sites that form a complex with AuCl_4_^−^ in the pernigraniline salt form PANI(PNS). After 1 min, the Au/N ratio indicates that only 0.5% imine sites are occupied by AuCl_4_^−^, and this value reaches almost 11.2% for 16 min exposure. Therefore, within the range of our experiment, a longer exposure time leads to a higher Au/N ratio.

When we compare the amount of Au after multiple deposition cycles in group C, the Au uptake is also linear and very similar to group B when the integrated Au exposure time throughout all the cycles is considered. Therefore, we can conclude that the amount of Au absorption in PANI per unit time is in a similar range for both processes. Furthermore, we calculated the Au concentration value of group C based on the measured NP size distributions and plot them together with the RBS data in Figure 10. While the calculated curve also indicates a linear increase, there is a concentration difference of around 9 × 10^15^ atoms/cm^2^ compared with the curve obtained via RBS. The calculated curve is obtained from NPs visible by the SEM on the PANI surface, whereas the RBS data measured the integrated Au concentration throughout the whole deposited PANI layer. Hence, it is assumed that there is still additional Au in PANI which may not have aggregated into Au NPs in excess of 5 nm or is buried below the surface and hence inaccessible by the SEM surface analysis.

### 3.5. Two-Step Growth Model

Based on our experimental results, we propose a model for the two-step growth of Au NPs in PANI thin film samples. According to the RBS results, the exposure of PANI with AuCl_4_^−^ at a voltage of +0.8 V vs. Ag/AgCl leads to a homogeneous distribution of Au throughout the PANI film and undesired reduction processes are unlikely. According to literature, the AuCl_4_^−^ can readily diffuse through PANI [26] and is attached to the protonated imine sites [12,27]. The nature of this interaction is not yet well understood, but the high affinity of AuCl_4_^−^ towards the imine is well-known [28,29]. It is evident from the experiments presented above that the reduction process leads to the formation of Au NPs. The reduction step, i.e., the lowering of the electrochemical potential on the PANI film, gives rise to two electrochemical conversion processes which cannot be distinguished by means of the CV analysis. On the one hand, the highly oxidized PANI(PNS) film is converted back to the emeraldine salt form PANI(ES), while on the other hand, the AuCl_4_^−^ is reduced to elemental Au. While the sequence and nature of these processes are unknown, we can discuss possible scenarios. The conversion to PANI(ES) means that the protonated imine sites transform into protonated amine sites which could lower the binding affinity of AuCl_4_^−^ which in turn gets liberated and therefore is able to diffuse through the PANI film. However, the PANI(ES) is also a good chemical reducing agent for AuCl_4_^−^ leading to Au(0) and consequently induces clustering and growth of Au NPs in line with other works that report on the electrochemical reduction of AuCl_4_^−^ to form Au NPs [29,30,31,32]. A different view is advocated by Janata and co-workers in their work on atomic metals [19,33]. Here, AuCl_4_^−^ is reduced at its initial imine site resulting in a single Au atom which becomes stabilized in the PANI framework. In the context of our work, the stabilization of single Au atoms by the PANI is unlikely due to the observation of many Au NPs and the apparent migration of Au to the PANI surface. We may explain this difference by the longer exposure time of AuCl_4_^−^ used in our work. According to our elemental analysis above, at 16 min exposure time compared to the 50 s used in the work of Janata, there is roughly one AuCl_4_^−^ per 10 imine sites. Considering the three-dimensional flexible framework of PANI, it becomes more likely that Au atoms cluster in the PANI at a higher Au concentration, destabilize and migrate to the surface.

The net result of Au NP formation is illustrated in Figure 11. We further illustrate in Figure 11 what we believe is happening in additional Au deposition cycles. Exposure of AuCl_4_^−^ will again lead to their homogeneous distribution throughout the PANI film. The reduction step leads the additional liberation of free Au species which tend to cluster to form new Au NPs as well as growing existing Au NPs. This scheme also provides an explanation for the multimodal size distribution of the Au NPs in samples with multiple deposition cycles.

We finally want to discuss briefly the stability of PANI(PNS)*AuCl_4_^−^ complexes in the absence of any stabilizing electrochemical potential. PANI(ES) is the only stable salt form under normal conditions, so any other PANI states will gradually change into this state and hence also the AuCl_4_^−^ will be reduced eventually. According to our experience, we cannot observe significant Au clustering in unreduced samples even after one week of storage under dry ambient conditions. However, as can be seen in Figure 4p, the exposure to the He ion beam generates a high density of finely dispersed Au NPs. The measurement spot can be easily spotted due to a substantial change of optical properties. From the fact that we observe much smaller Au NPs in this case, we can conclude that the reduction in the acid solution enhances the clustering processes of the Au NPs.

## 4. Conclusions

In this work, we investigated the deposition of Au into PANI by a two-step electrochemical process consisting of AuCl_4_^−^ exposure at a constant electrochemical potential and a controlled reduction by means of CV following the rinsing of excess AuCl_4_^−^. The SEM, TEM and further RBS analysis clearly demonstrate that the AuCl_4_^−^ is homogeneously dispersed in the PANI during the exposure step and the following reduction step leads to Au NP clustering and surface segregation. Moreover, multiple cycles of deposition lead to polydispersed Au NP distributions due to the fact the previously formed Au NPs grow as well as new Au NPs are formed. In previous works, this method was used to prepare Au clusters with precisely defined numbers of atoms in PANI. This could not be realized in our work as the precursor doses we used were probably too high. It will be interesting to perform a more detailed analysis of the unreduced samples by means of higher resolution methods to gain further insight into the PANI(PNS)*AuCl_4_^−^ complexation.

## Figures and Tables

**Figure 1 nanomaterials-13-02089-f001:**
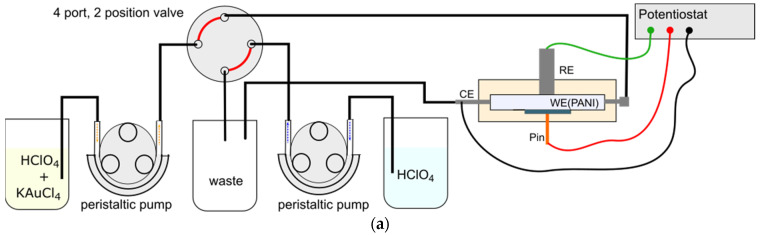
Description and principle of the Au deposition process: (**a**) schematic drawing of experimental setup where the designed flow cell is used as an electrochemical cell; (**b**) cross-section of the designed flow cell for the Au deposition process where continuous electrochemical contact is maintained; and (**c**) schematic diagram of the Au deposition process from Stage I to Stage V with variations of different salt states of PANI. The symbol * represents the attachment of AuCl_4_^−^ to PANI(PNS).

**Figure 2 nanomaterials-13-02089-f002:**
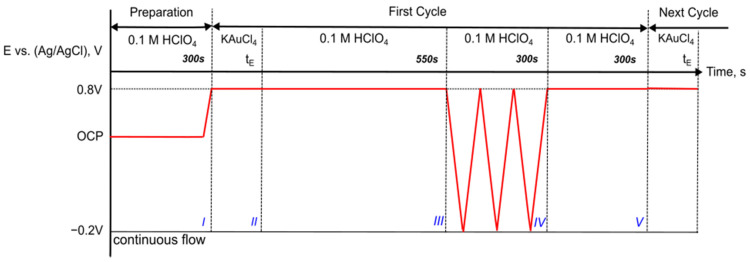
Timing diagram of Au deposition in PANI with corresponding states shown in Figure 1c.

**Figure 3 nanomaterials-13-02089-f003:**
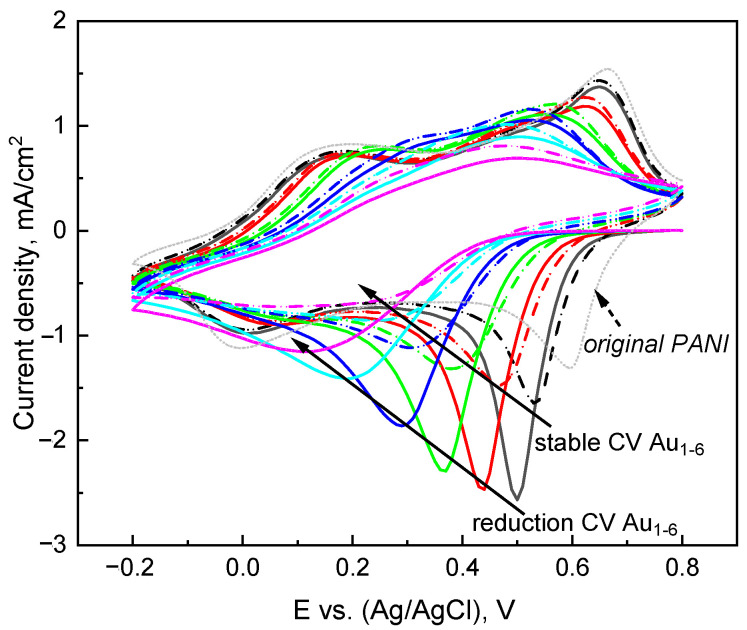
Overlay of each individual CV of a PANI/Au_6_ sample with corresponding first reduction sweep after holding at +0.8 V and subsequent stable CV with 1 min AuCl_4_^−^ exposure time. The original PANI CV is plotted for comparison. The different colors represent the CV of Au_1_ to Au_6_.

**Figure 4 nanomaterials-13-02089-f004:**
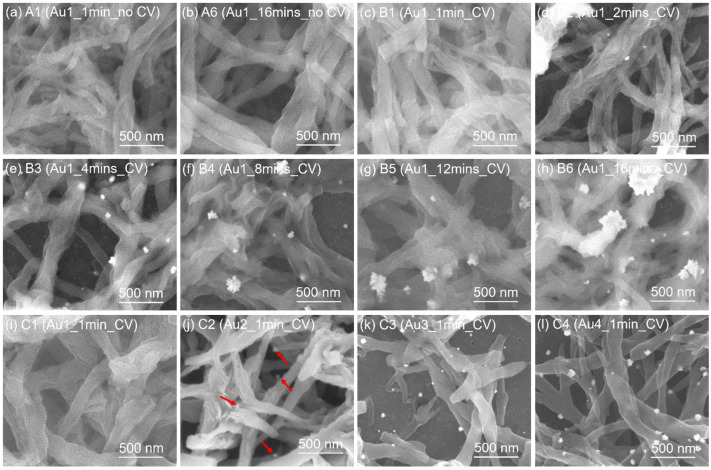
SEM images of PANI/Au samples with different experimental parameters: (**a**) A1; (**b**) A6; (**c**) B1; (**d**) B2; (**e**) B3; (**f**) B4; (**g**) B5; (**h**) B6; (**i**) C1; (**j**) C2 where the arrows indicate the appearance of Au NPs; (**k**) C3; (**l**) C4; (**m**) C5; (**n**) C6; (**o**) D; and (**p**) A6 after RBS.

**Figure 5 nanomaterials-13-02089-f005:**
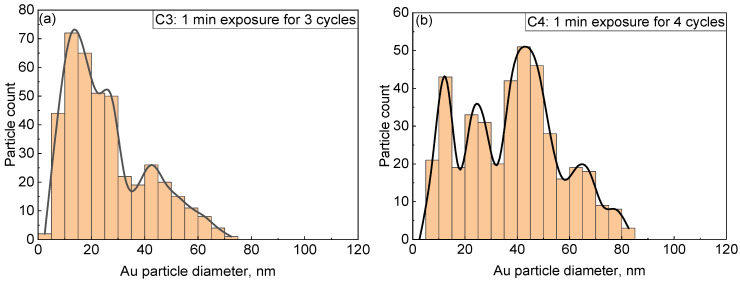
NP size distributions of PANI/Au from group C: (**a**) C3 of Au [3c, 1 min, CV]; (**b**) C4 of Au [4c, 1 min, CV]; (**c**) C5 of Au [5c, 1 min, CV]; and (**d**) C6 of Au [6c, 1 min, CV].

**Figure 6 nanomaterials-13-02089-f006:**
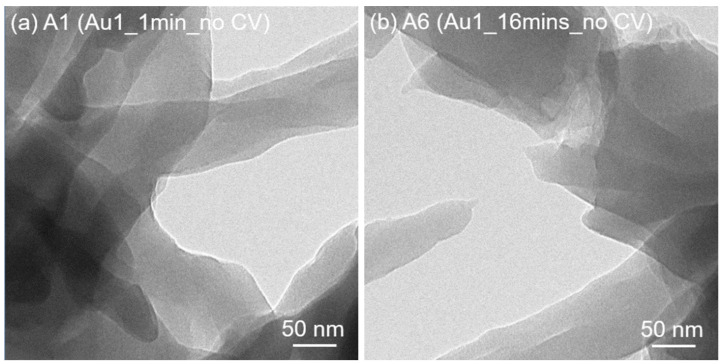
TEM images of PANI/Au samples with different experimental parameters according to Table 1: (**a**) A1 of Au [1c, 1 min, no CV]; (**b**) A6 of Au [1c, 16 min, no CV]; (**c**) B1 of Au [1c, 1 min, CV]; (**d**) B6 of Au [1c, 16 min, CV]; and (**e**) HRTEM and FFT of B6.

**Figure 7 nanomaterials-13-02089-f007:**
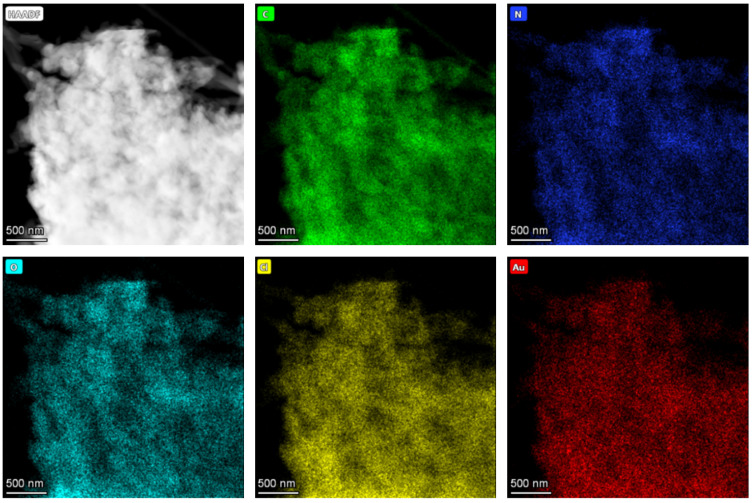
HAADF-STEM image of Au [1c, 16 min, no CV] where no NPs are found and the corresponding EDS mapping.

**Figure 8 nanomaterials-13-02089-f008:**
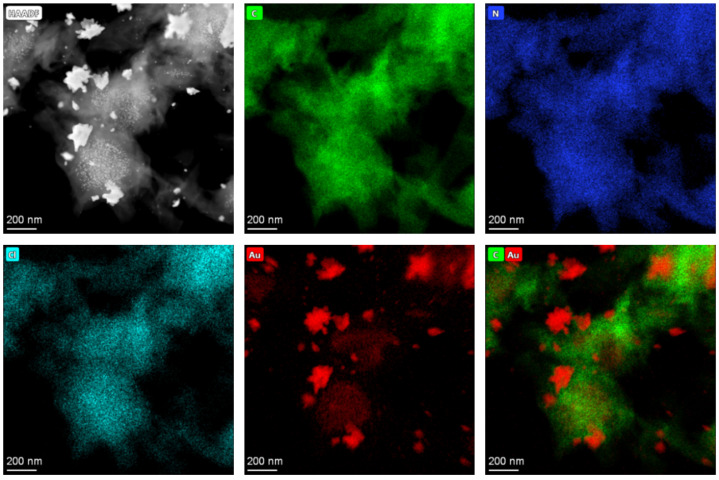
HAADF-STEM image of Au [1c, 16 min, CV] with NPs and the corresponding EDS mapping.

**Figure 9 nanomaterials-13-02089-f009:**
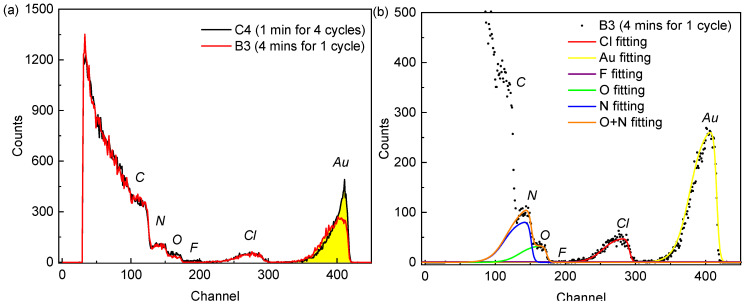
RBS spectra of PANI/Au according to Table 1: (**a**) spectra of Au [1c, 4 min, CV] and Au [4c, 1 min, CV], and (**b**) fitting results of measured element peaks in Au [1c, 4 min, CV] by the routine described in the text.

**Figure 10 nanomaterials-13-02089-f010:**
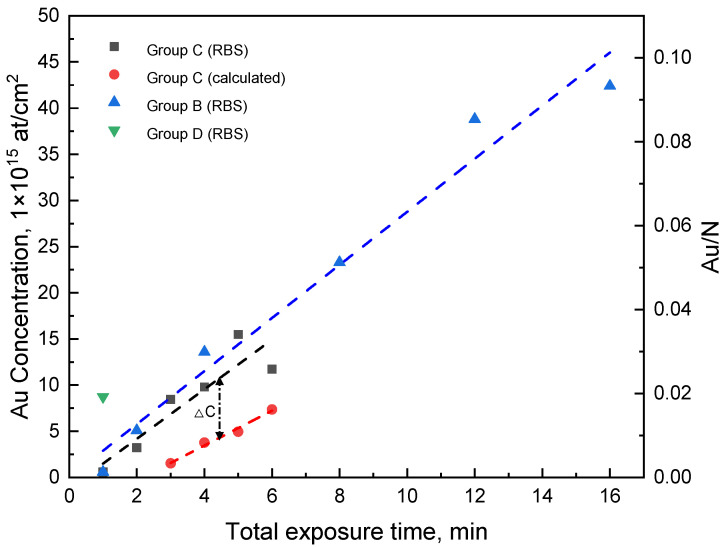
The Au concentration of deposited PANI in different experimental groups. The red dashed curve is the calculated Au concentration value based on measured NP size distributions in Figure 5.

**Figure 11 nanomaterials-13-02089-f011:**
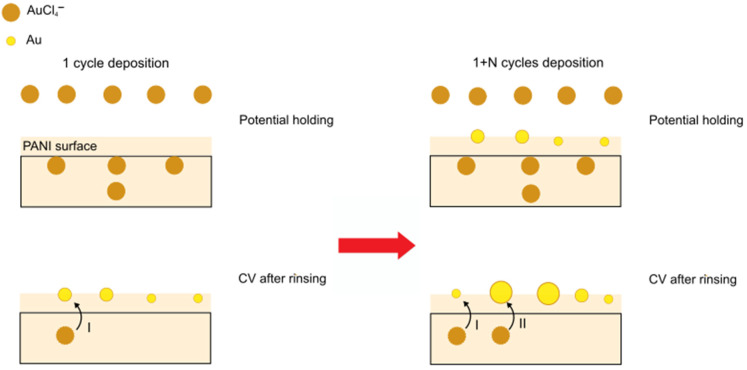
Two-step growth model of a single Au deposition cycle and the effect of additional deposition cycles. At first, AuCl_4_^−^ anions are dispersed within the PANI structure and controlled reduction leads to the formation of Au NPs. Multiple steps lead to the formation of new Au NPs (I) as well as to the growth of already existent Au NPs (II).

**Table 1 nanomaterials-13-02089-t001:** Experimental parameters of PANI Au deposition.

Sample	Number of Cycles, N	KAuCl_4_ Concentration, mM	Exposure Time, min	CV
A1	1	0.1	1	No
A2	1	0.1	2	No
A3	1	0.1	4	No
A4	1	0.1	8	No
A5	1	0.1	12	No
A6	1	0.1	16	No
B1	1	0.1	1	Yes
B2	1	0.1	2	Yes
B3	1	0.1	4	Yes
B4	1	0.1	8	Yes
B5	1	0.1	12	Yes
B6	1	0.1	16	Yes
C1	1	0.1	1	Yes
C2	2	0.1	1	Yes
C3	3	0.1	1	Yes
C4	4	0.1	1	Yes
C5	5	0.1	1	Yes
C6	6	0.1	1	Yes
D	1	1	1	Yes

## Data Availability

Data available on request.

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
