# Peer review of "Two-Step Electrochemical Au Nanoparticle Formation in Polyaniline"

_nanomaterials, 2023, doi:10.3390/nano13142089_

Round 1
Reviewer 1 Report
The manuscript by Zhao et al. reported a route to insert Au nanoparticles in polyaniline via two step electrochemical method. The materials were characterized by Transmission electron microscopy (TEM), energy dispersive X-ray spectroscopy (EDS) and Rutherford backscattering spectrometry (RBS). The research topic and the presented data are interesting and match the scope of the Nanomaterials journal. It can be accepted after addressing the following minor comment:
> Author should include the XRD pattern of the prepared samples and discuss the findings in the main text of the manuscripts.
> Comment on the novelty of this method over other methods for the integration of Au nanoparticles in the PANI matrix.
Reviewer 2 Report
In this paper, authors have developed and optimized an electrochemical methodology to deposit gold nanoclusters/gold nanoparticles on top of polyaniline electrodes. The electrodes have been mainly characterized with microscopy analysis. The text is too descriptive and some paragraphs present low quality of scientific discussion. Authors must include more characterization techniques that support their observation. Despite of the paper could be interesting for the Nanomaterials MDPI audience; it should be improved first remarkably.
- - Authors should improve the experimental section, detailing better the equipment used, sample preparation etc
- - In Table 1, the authors should detail properly the way that they have calculated the Au concentrations. Moreover, Table 1 with all the information appears on page 4, and the calculation way to measure the amount of gold in page 9. The authors should improve the dialog line of the paper.
- Authors claim: “It indicates that Au was initially well dispersed in PANI, but the high He ion energy resulted in the formation of Au NPs significantly smaller than CV reduction in solution”. Could the authors develop this more in detail this hypothesis?
- In order to improve the manuscript, I would encourage the authors to change the type of name A or B for a name or description that evokes better the idea of the experiment. Authors should improve also the descriptive way of showing the results. There are no connections between the descriptions of the results or enough cites to support their observations
- Authors are claiming this “… we can estimate a “compacted” PANI layer thickness of ~540 nm assuming a literature value of the volume density of PANI of 1.329 g/cm³ [25]…”. The experiments that are considered might be forming different types of films, with different types of thickness. This assumption should be corroborated by the authors or look for another way to calculate compaction/surface covered or volume of the film
- As well, I would encourage the authors the use XPS for the quantification of gold in the sample. I believe it could improve the quality and accuracy of the paper.
- I would encourage the authors to show the CVs of the PANI formation, as well as the reduction of the gold on the surface. The CVs should change drastically, or some reduction peaks should also appear, especially depending on the type of gold obtained. As an example of different CVs in Au-type function: https://iopscience.iop.org/article/10.1088/1742-6596/307/1/012061, https://www.sciencedirect.com/science/article/abs/pii/S0925400518321270
- XPS could also assist and improve the hypothesis made in the two-step growth model. Authors should discuss and improve the hypothesis, assisting with a comparison with others reported in the literature on electrochemical growth mechanisms method of growth.
